# Preliminary Results on the Comparative Evaluation of Alkaline Phosphatase Commercial Tests Efficiency in Non-Cow Milk Pasteurization

**DOI:** 10.3390/biotech11030039

**Published:** 2022-08-26

**Authors:** Asimo Tsiamita, George Valiakos, Nikolaos Natsaridis, Stamatia Fotiadou, Athanasios Manouras, Eleni Malissiova

**Affiliations:** 1Department of Medicine, University of Thessaly, 41500 Larissa, Greece; 2Faculty of Veterinary Science, University of Thessaly, 43100 Karditsa, Greece; 3Food of Animal Origin Laboratory, Animal Science Department, University of Thessaly, 41500 Larissa, Greece; 4Independent Researcher, 60100 Katerini, Greece; 5Nutrition and Dietetics Department, University of Thessaly, 43100 Karditsa, Greece

**Keywords:** alkaline phosphatase, pasteurization, goat milk, sheep milk, buffalo milk, camel milk, donkey milk, total viable count, Enterobacteriaceae

## Abstract

The demand for non-cow milk and the products derived from it, is constantly increasing; thus, correct and effective pasteurization becomes necessary. Typical practices for evaluating milk pasteurization are mainly based on the thermal inactivation of an endogenous enzyme, alkaline phosphatase (ALP). The ALP tests, originally designed and applied to pasteurized cow milk, are often used to control pasteurization in non-cow milk, without sufficient data on their suitability; EFSA calls on the scientific world for collecting more information on the subject. In this study, the pertinent details of the ALP assay for non-cow milk products are summarized, and a comparison is performed regarding the evaluation of the adequacy of commercially available tests for the determination of ALP activity in non-cow milk. At the same time, raw and pasteurized non-cow milk was analyzed microbiologically using standard ISO methods and MALDI-TOF MS in order to confirm the thermal effect on common microorganisms. In these preliminary results, various ALP tests do not appear to be fully reliable as indicators for the pasteurization of some types of non-cow milk such as camel and donkey milk or even goat and sheep milk, using the EFSA proposed limits. ALP commercial kits may not be suitable as pasteurization indicators for various types on non-cow milk, and alternatives should be investigated.

## 1. Introduction

It is well established nowadays that pasteurization of raw milk, colostrum, dairy or colostrum-based products must be achieved using heat treatment, such that the alkaline phosphatase (ALP) test immediately after such treatment gives a negative result [1]. However, it is a major concern that the ALP test was originally designed and applied to pasteurized cow milk; hence there is insufficient data available with respect to the use of this test for evaluating pasteurization in other animal species milk. The currently available data on the subject for the various types of non-cow milk are as follows:

### 1.1. Goat and Sheep Milk

In general, goat milk shows lower levels of ALP activity, up to five times, compared to that of cow milk [1,2]. In sheep milk, on the other hand, ALP is more active, two to three times more than cow milk, and increases during lactation [3]. According to Raynal-Ljutovac et al. [4], there are significant differences in ALP content between species, breeds within species, and individual animals. Studies have shown that the highest percentage of ALP was present in buttermilk, and when the cream was stirred for milk production, half of the amount observed in the cream penetrated in the produced milk [5]. In a study performed by Vamvakaki et al. [6], during milk heat treatment, ALP activity showed a faster declining trend in goat and sheep milk in contrast to cow milk, which showed a slow inactivation of ALP. Wilińska et al. in 2007 [7] also showed that the stability of ALP in cow milk was much higher than that in goat and sheep milk. The temperature range that was used to heat the samples was higher, ranging from 54–69 °C for 1–180 min [1,7].

The same result was achieved by Dumitraşcu et al. in 2014 [8] who used a fluorometric method in whole and skimmed milk. In whole milk at low temperatures, the ALP residue was higher in goat milk due to more fat. As the temperature increased, the residuals in goat milk decreased, while in cow milk they remained stable. Regarding skimmed milk, goat milk showed the lowest ALP activity, followed by sheep and lastly cow milk, in which almost twice the amount was found compared to the other two species. So, the conclusion they came to is that ALP in cow milk is more tolerant and resistant of temperature rise [1,8].

Klotz and his colleagues studied and attempted to evaluate the Fluorophos method and the MFO-3 colorimetric analysis for the detection of ALP in a fresh and pasteurized milk sample. The results of their research showed that in all three types of milk (goat, sheep and cow) the reduction of the enzyme activity is dramatic between the 67.0 and 72.5 °C pilot pasteurization trials [9].

Based on the above, it is generally demonstrated that there is a slower decrease in ALP activity in cow milk compared to that in goat and sheep milk, while the latter had the fastest decrease in ALP activity. Sensitivity of the two methods for detecting ALP activity has not been determined. In various studies, the level of ALP in milk varies widely depending on the season, the breed of the animal and the stage of lactation. Goat and sheep milk naturally contain ALP at about 10 and 200%, respectively, of the level in cow milk. At present, the Canadian maximum level of residual ALP in pasteurized goat and sheep milk is the same as that of cow milk. Due to fluctuations in baseline ALP levels, it has been proposed to set different legal limits for milk from different species to verify the adequacy of pasteurization [3].

In addition, some colorimetric methods are less sensitive and have been shown to have a significant failure rate. In order to increase the sensitivity of colorimetric assays for non-cow milk, Williams and Nottingham [10] modified the Aschaffenburg and Mullen assay process by increasing the sample volume. The application in goat milk proved to be satisfactory. Furthermore, ALP values in goat milk have been reported to vary considerably, with little or no correlation between milk fat content and milk solids. Furthermore, it has been observed that in cases of mastitis, milk has a higher activity of ALP [11].

Barbosa considered that differences in the fat content of both goat and sheep milk used for direct human consumption or cheese production influenced the performance of the available ALP tests [12].

Moreover, in studies carried out by Istituto Zooprofilattico Sperimentale Lazio e Toscana [3], the variability of the limits for the correct pasteurization of each species was demonstrated and how they should be adjusted were suggested. Thus, based on the data at their disposal, they concluded that to have the goat milk sufficiently pasteurized the ALP levels must be below the 330 mU/L limit, while for sheep milk, below 530 mU/L. Similar limits are being proposed by European Food Safety Authority (EFSA) (300 and 500 mU/L, respectively) but it is recommended that further data needs to be collected in order to conclude if these limits are valid of all situations [1].

### 1.2. Camel Milk

Studies on the evaluation of ALP activity in camel milk, both raw and pasteurized, are limited. One of these was conducted by Lorenzen et al., in 2011 [13]. They observed that ALP values in pasteurized milk did not differ much from those found in raw milk. It is worth mentioning that the values were measured by fluorometric method and by chromatography. The conclusion was that ALP is not the appropriate enzyme to verify the proper pasteurization of camel milk due to increased residual activity [1,13]. Wernery’s scientific team reached the same conclusion [14]. They used a total of four methods to determine adequate pasteurization, one fluorometric, one photometric and two chromatographic. They observed that the ALP reduction rate after several hours of heat-treated milk was minimal and that the enzyme was almost not inactivated after heating to 72 °C, which is the acceptable High Temperature Short Time (HTST) pasteurization temperature for other species milk [1,14].

### 1.3. Buffalo Milk

One of the relevant studies regarding buffalo milk showed that the ALP activity in buffalo milk is not particularly related to the activity of the corresponding enzyme in cow milk [15]. Nevertheless, the degree of enzyme reactivation in the two types of milk is similar. In 2000, Lombardi’s team found [16] that ALP showed the highest sensitivity (vs three other milk enzymes measured) to heat inactivation at 60 °C, while at 70 °C, the enzyme was completely inactivated in one minute. In addition, the activity of the enzyme did not seem to differ much between buffalo and cow milk, while it is concentrated in both cases in the creamy phase. A study was conducted by the International Dairy Federation and the International Organization for Standardization to evaluate the reproducibility of a chemiluminescence method for ALP at 50, 100, 350 and 500 mU/L in whole milk of multiple species. The results of this study revealed that this method was comparable to fluorescence analyses and indicated that the chemiluminescence method is suitable for measuring ALP in milk derived from multiple species and in dairy drinks in the US and EU at levels below 350 mU/L [17].

### 1.4. Donkey Milk

In 2009, Marchand et al. [18] examined the ALP contained in horse milk, which also displays a significant resemblance to donkey milk. The aim of this study was to demonstrate the ALP potential to be used as an enzymatic marker in milk. Although, from a kinetic point of view, the enzyme can be used as an indicator for the proper pasteurization of milk, nevertheless it cannot be utilized due to the low endogenous level of enzyme present in horse milk, and probably in donkey milk. Specifically, for the assessment of ALP in raw horse milk, fractionation was performed by centrifugation. The results from horse milk, compared to cow milk, showed that the activity of ALP was much lower, as well as the distribution of enzymes in milk was completely different [18]. This is most likely due to the lower percentage of fat found in this type of milk.

Our team performed a preliminary evaluation of the efficiency of commercial ALP test kits to determine possible suitability for use in non-cow milk pasteurization. At the same time, raw and pasteurized non-cow milk was analyzed microbiologically using standard ISO methods and MALDI-TOF MS in order to confirm the thermal effect on common microorganisms.

## 2. Materials and Methods

*Samples collection*: A total of 10 samples of raw milk, 500 mL each, were collected in sterile containers, from sheep, goat, buffalo, donkey and camel farms in Greece and Cyprus. The samples were partitioned into analytical samples of 50 mL and stored under freezing conditions (−25 °C) until analysis. After controlled thawing, their pH was checked to determine their suitability for further analysis.

*Samples processing*: All raw samples (10) were pasteurized with HTST method (72 °C, 15 s) and Low Temperature Long Time (LTLT) method (63 °C, 30 min) in a semi-industrial environment. During the pasteurization, 200 mL of the pasteurized samples by the HTST method was placed in Erlenmeyer flasks. The heating took place in a water bath, while the temperature was constantly monitored with a thermometer. For LTLT pasteurization, 200 mL of each raw milk sample was placed in metal containers and then heated on heating plates, for 30 min at 63 °C. The heat-treated samples (20) were stored under refrigeration or freezing for further analysis.

*Microbiological analysis*: A total of 30 samples (raw and pasteurized) were analyzed for Total Viable Count (TVC) and Enterobacteriaceae using standard methods (ISO 4833-1:2013, ISO 21528-2:2017 respectively). Samples were prepared with initial suspensions and decimal dilutions for microbiological examination according to ISO 8261:2001. TVC was enumerated on Plate Count Agar (PCA) incubated at 30 °C for 72 h. Enterobacteriaceae were enumerated on Red Bile Glucose (VRBG) Agar incubated at 37 °C for 24 h. All samples were analyzed in duplicate. Enterobacteriaceae isolates were identified at species level by Matrix-Assisted Laser Desorption/Ionization-Time of Flight Mass Spectrometry (MALDI-TOF MS), using a Microflex LT (Bruker Daltonik GmbH, Bremen, Germany) mass spectrometer. A single colony from freshly grown isolates was picked and directly applied to a steel MALDI target plate. Afterwards, it was overlaid with one µL of a saturated solution of α-cyano-4-hydroxycinnamic acid (HCCA) (Bruker Daltonics, Bremen, Germany) and allowed to co-crystalize at room temperature. Spectra were automatically acquired in a linear positive mode, at a laser frequency of 20 Hz, within a mass range from 2000 to 20,000 Da, with AutoXecute acquisition software (Flex control 3.4, Bruker Daltonics, Bremen, Germany). The spectra were externally calibrated using Escherichia coli DH5alpha. Raw spectra were processed using MALDI BioTyper v.3.1 software (Bruker Daltonik GmbH, Bremen, Germany). Results were classified using modified score values proposed by the manufacturer.

*Determination of alkaline phosphatase activity*: All 30 samples, namely 10 raw, 10 pasteurized with the HTST method and 10 pasteurized with the LTLT method were tested for ALP activity with three qualitative tests (A, B and C) and three quantitative tests (D, E and F) commercially available and randomly selected. All tests were performed following strictly the manufacturer’s instructions, and all samples were analyzed in duplicate. As the activity of ALP in buffalo milk is similar to cow milk, this type of milk served as a positive control for all tests.

The ALP Qualitative Tests used were Lactognost (HEYL, Berlin, Deutschland), Phosphatesmo (MACHEREY-NAGEL, Dueren, Germany) and Lactopast Biomedix (MenidiMedica, Menidi, Greece) hereafter A, B and C respectively. For the technical characteristics of each test, briefly refer to the following: Lactognost is a qualitatitive test that uses three reagents, namely Lactognost I (buffer solution), II (disodium phenyl phosphate) and III (chloro imino dibromo quinine). A tablet of Lactognost I and II was dissolved in 10 mL of water and 1 mL of the milk sample was added. Incubation at 37 °C for 1 h followed, and at the end Lactognost III was added. Exactly 10 min later, the color was evaluated: Brown corresponds to ALP Negative (pasteurized milk), Blue to ALP Positive (unpasteurized milk) and Green to ALP traces (insufficiently pasteurized milk). Phosphatesmo is a test strip that was briefly dipped into the milk sample. The excess of liquid milk was shaken off, and in order to prevent the test pad of the test strip from drying out, the test strip was put in the provided by the kit bag. Samples were then incubated at 36 °C for 1 h. In the presence of ALP, the test field turns yellow. A yellow coloration indicates that raw milk is present, or the milk was not sufficiently heated. No coloration indicates that the pasteurization was correctly completed. Lactopast Biomedix (MenidiMedica uses a total of 400 μL of R2 reagent that was first placed in each tube, followed by 100 μL of R3 reagent. Then 10 μL of milk was added, and the test tube was shaken for 5 s. In the next 5 s, the analysis was complete. If the color remains the same, then pasteurization has been achieved. Otherwise, if the sample turns to yellow or green, then this is an indication of incorrect pasteurization.

The ALP Quantitative Tests used were ZymoSnap ALP (Hygiena, Huntingdon, UK), PasLite test (Charm Sciences, Lawrence, KS, USA) and Fluorophos (Advanced Instruments, Norwood, MA, USA), hereafter D, E and F, respectively. For the technical characteristics of each test, briefly refer to the following: ZymoSnap ALP principle is based on the use of a single stand-alone device, in a simple procedure. The milk sample is placed in the ZymoSnap ALP tube, the device is activated to release the detection reagent, and the sample is incubated for 5 min. ALP activity is measured in the EnSURE monitoring system. The results were displayed in 15 s. The device was calibrated using the ZymoSnap ALP Positive Control Kit. For using PasLite test, during the preparation of the dairy samples to be analyzed, they are mixed with PasLite reagents and incubated at 35 °C for 3 min. The resulting solution emits radiation in visible light, intensity directly proportional to the concentration of the enzyme in it. The Charm novaLUM II-X system is used to measure the emitted light and converts the light measurements into enzyme units. The results appeared after 5 s. The measurement of all tubes was completed within 3 min from the addition of the Stopping Solution. For implementing Fluorophos, a total of two mL of the reconstituted Fluorophos^®^ ALP Substrate was dispensed into labeled fluorometer cuvettes for each test. The cuvettes were then placed into the heating block and incubated for 15 min at 38 ± 1 °C. An aliquot of 75 μL of the sample was placed in the preheated cuvette and mixed well with a vortex mixer. The cuvette was placed in the fluorometer cuvette chamber. After 60 s, the fluorometer began to measure and displayed the fluorescence of the sample in fluorescence units (FLU). After three minutes, the fluorometer displayed the average increase in fluorescence and the ALP activity in mU/L or mU/kg.

## 3. Results

### 3.1. Microbiological Results

Table 1 shows the results regarding the Total Viable Count (TVC) and enumeration of Enterobacteriaceae. The results show that the highest TVC was found in goat and sheep, while the lowest was found in donkey milk, before and after pasteurization. Regarding Enterobacteriaceae, the highest numbers were detected in sheep, goat and buffalo milk, but after pasteurization, members of the family were not detected in any of the samples.

Regarding Enterobacteriaceae identification at species level by MALDI-TOF MS, the most common microorganisms found in raw milk samples are presented in Table 2.

### 3.2. Qualitative Alkaline Phosphatase Test Results

Table 3 shows the results of the qualitative ALP tests used in this study. Specifically, it is reported whether the results related to ALP are positive (+), negative (-), or doubtful (~).

### 3.3. Quantitative Alkaline Phosphatase Test Results

Table 4 shows the results of the quantitative ALP tests used in this study. Specifically, it is reported whether the results related to ALP are positive (+) or negative (-) depending on the results received: the known cow’s milk limit of >350 is declared as + and not considered pasteurized, while values < 350 are considered negative, except goat and sheep milk where values of 300 and 500 mU/L, respectively, were considered as limits, as proposed by EFSA.

Table 5 shows the comparison of the results of all tests used in the study for all 30 milk samples.

## 4. Discussion

The present study was carried out in order to comparatively evaluate the reliability of commercial ALP tests for non-cow milk, utilizing three qualitative and three quantitative kits, randomly selected, which were used for the selected samples of goat, sheep, camel, donkey and buffalo milk. The kits used are the most commonly used in the dairy industry and are therefore considered representative. The samples covered important non-cow milk types consumed worldwide. The analyses of the samples were completed in duplicate for reasons of reliability of the results.

Regarding the heat treatment of the samples, this was performed in laboratory conditions, and there may be deviations from its industrial application. Nevertheless, the conditions chosen simulate the time–temperature relationships used both in the dairy industry and in small cheese factories and are therefore considered to be representative for the sanitization of milk. Nevertheless, in order to have a safety valve for the successful heat treatment, the samples were tested microbiologically before and after the treatment to establish its effect.

During the microbiological control of milk samples, both fresh and pasteurized, a decrease in both TVC and the populations of Enterobacteriaceae was found, which was expected and may indirectly indicate the effectiveness of the heat treatment.

All three qualitative (color) tests examined showed discrepancy in the results. More specifically, test A recognized all raw milk samples as non-pasteurized but did not recognize five of the samples pasteurized with the HTST method (both goat and camel milk samples and one sheep milk sample) and six of the samples pasteurized with the LTLT method (both goat, camel and donkey samples). Both B and C tests recognized all pasteurized milk samples with both methods but failed to recognize some raw milk samples as non-pasteurized: three raw samples for the B test (one goat and both donkey samples) and the same samples plus one of the camel milk samples for the C test. In total, donkey, goat and camel milk samples gave the most discrepant results. Especially concerning for the public health is the failure to detect ALP activity in not sufficiently pasteurized or raw milk, as in the cases of these three types of milk in our study. Consequently, these test kits for non-cow milk should be performed with caution, since they may contribute to the characterization of unsuitable samples as safe for the consumer, especially in the case of donkey, camel and goat milk.

Regarding kits that perform quantitative determination, it is noted that acceptable values for pasteurized samples should be lower than 350 mU/L, except goat and sheep milk where values of 300 and 500 mU/L, respectively, were considered the limits in our study, as proposed by EFSA. From the three tests, test E gave the most accurate results, even though the values estimated in this kit were the lowest compared to the other tests. However, it could not detect the raw donkey milk as non-pasteurized. The D and F tests had difficulty recognizing HTST sheep and goat milk (one of the samples in each case). The greatest issue was demonstrated regarding the camel samples: test E was the only one that could recognize HTST and LTLT pasteurization of this type of milk. Both D and F tests showed a high ALP activity in both camel milk samples and in both pasteurization methods.

The buffalo milk seems to respond satisfactorily with all the tests, qualitatively and quantitatively. It was the only type of milk with completely valid results in all tests; as reported before, the activity of ALP in buffalo milk is similar to cow milk. According to Lombardi et al. [16], buffalo milk ALP exhibits maximal inactivation sensitivity at 60 °C and is completely inactivated at 70 °C.

Regarding camel milk, this study demonstrates that most commercial ALP kits cannot be used as pasteurization indicators. As reported in the introduction, previous studies demonstrated that the ALP reduction rate after several hours of heat-treating camel milk was minimal and that the enzyme was almost not inactivated. In our study, only the B qualitative test and the E quantitative test gave accurate results in both camel milk samples. Regarding camel milk, the scientific team of Wernery et al., in 2008 [14], also found that ALP could not be a reliable indicator of its pasteurization. This is due to the higher temperature required for the total deactivation of the enzyme compared to that applied during the heat treatment of milk. They suggest, as an alternative, the study of other enzymes, such as galactoperoxidase (POD) or γ-glutamyl transferase, as they seem to be more valid. This was also observed in the experiments conducted in our laboratory with the various commercial kits. In the camel milk samples analyzed quantitatively, ALP values were elevated in D and F tests. Moreever, of the qualitative determination kits, camel milk is not shown to be sufficiently pasteurized by the A test for both the HTST and LTLT pasteurized samples.

As we have described in a previous section, horse milk has, in general, the same characteristics as donkey milk, with similar activity of ALP enzymes contained in both. In a study by Marchand et al. [18] on the kinetics of ALP in horse milk, it was shown that its ALP cannot be a reliable indicator of pasteurization of said milk. Furthermore, low levels (<0.2%) of raw milk contamination in the pasteurized product cannot be detected due to these low endogenous levels of equine ALP and the consequent detection limitations of the method. Therefore, even though it is considered pathogen-free milk, the use of ALP as an indicator cannot be guaranteed. However, it could be exploited if the reference method was 200-fold more sensitive (e.g., by increasing the incubation time and/or using more sensitive substrates) [18]. Malissiova et al. [19], reported that it seems that ALP cannot be considered as a valid indicator for donkey milk safety in relation to pathogens as it becomes inactivated in lower temperature and time combinations in comparison to the classical pasteurization conditions.

In agreement with the above, selected qualitative and quantitative determination kits in the present study were not found to be suitable for donkey milk, showing it as free of ALP, even when it had not undergone pasteurization. Quantitative Tests D and F were the only ones with valid results in all scenarios.

Concerns arise regarding the use of ALP commercial kits in sheep and goat milk as well. Regarding goat milk, all qualitative tests had at least one false result, the most concerning being the inability to recognize raw goat milk as non-pasteurized. In quantitative tests, the results were more acceptable with only one false positive result near the limit in an LTLT goat sample. In sheep milk samples, qualitative tests were more accurate with only one false doubtful result in an HTST-treated sample. Regarding quantitative results, two false positive results on HTST-treated samples add more concerns with respect to whether the rise of the limit to 500 mu/L for sheep milk as proposed by EFSA is enough, or more data should be collected from more sheep milk samples worldwide to evaluate the use of ALP activity as a pasteurization indicator in this type of non-cow milk.

In conclusion, the above available ALP activity detection kits as it has been shown through these preliminary data are not suitable for all milk species. As proposed by EFSA and the scientific community [1,20], it is important to carry out further studies and collect more data on various species types of milk in order to evaluate the validity of various ALP limits or the validity of some other enzyme indicators more suitable for indicating appropriate pasteurization in non-cow milk.

## Figures and Tables

**Table 1 biotech-11-00039-t001:** Results of TVC and Enterobacteriaceae (cfu/mL).

Sample	Raw Milk (cfu/mL)		HTST (cfu/mL)		LTLT (cfu/mL)	
	TVC	Enterobacteriaceae	TVC	Enterobacteriaceae	TVC	Enterobacteriaceae
Goat milk S1	>3 × 10^6^	4.55 × 10^3^	5.82 × 10^4^	0	8.73 × 10^3^	0
Goat milk S2	>3 × 10^6^	8.55 × 10^3^	988	0	13 × 10^5^	0
Sheep milk S1	>3 × 10^6^	1.56 × 10^4^	122	0	400	0
Sheep milk S2	>3 × 10^6^	1.14 × 10^4^	1.89 × 10^3^	0	346	0
Camel milk S1	6.21 × 10^4^	0	3.46 × 10^3^	0	2976	0
Camel milk S2	3 × 10^4^	0	1187	0	18	0
Donkey milk S1	5.09 × 10^3^	10	124	0	254	0
Donkey milk S2	5.73 × 10^3^	8	432	0	4	0
Buffalo milk S1	2.35 × 10^5^	4.36 × 10^3^	3.5 × 10^4^	0	12.25 × 10^4^	0
Buffalo milk S2	2.73 × 10^5^	3.64 × 10^3^	4.09 × 10^4^	0	967	0

**Table 2 biotech-11-00039-t002:** Results of MALDI-TOF MS detection in raw milk samples.

	Microorganisms
Goat milk S1	*Pantoea agglomerans*
Goat milk S2	*Enterobacter cloacae*, *Pantoea agglomerans*, *Klebsiella oxytoca*, *Escherichia coli*
Sheep milk S1	*Pantoea agglomerans*, *Hafnia alvei*
Sheep milk S2	*Klebsiella oxytoca*, *Pantoea agglomerans*
Camel milk S1	*None*
Camel milk S2	*None*
Donkey milk S1	*Klebsiella oxytoca*, *Pantoea agglomerans*, *Enterobacter cloacae*
Donkey milk S2	*Enterobacter cloacae*
Buffalo milk S1	*Pseudomonas fulva*, *Pseudomonas koreensis*, *Klebsiella oxytoca*, *Pantoea agglomerans*
Buffalo milk S2	*Pseudomonas koreensis*

**Table 3 biotech-11-00039-t003:** Results of qualitative alkaline phosphatase tests for non-cow milk samples.

	Raw Milk			HTST			LTLT		
ALP Test	A	B	C	A	B	C	A	B	C
Goat milk S1	+	- *	- *	~ *	-	-	~ *	-	-
Goat milk S2	+	+	+	~ *	-	-	~ *	-	-
Sheep milk S1	+	+	+	~ *	-	-	-	-	-
Sheep milk S2	+	+	+	-	-	-	-	-	-
Camel milk S1	+	+	+	+ *	-	-	+ *	-	-
Camel milk S2	+	+	- *	+ *	-	-	+ *	-	-
Donkey milk S1	+	- *	- *	-	-	-	~ *	-	-
Donkey milk S2	+	- *	- *	-	-	-	~ *	-	-
Buffalo milk S1	+	+	+	-	-	-	-	-	-
Buffalo milk S2	+	+	+	-	-	-	-	-	-

* results that are considered not correct, according to milk sample status

**Table 4 biotech-11-00039-t004:** Results of quantitative alkaline phosphatase tests (mU/L) of non-cow milk samples.

	Raw Milk						HTST						LTLT					
ALP Test	D		E		F		D		E		F		F		E		F	
**Goat milk S1**	**+**	**11557.5**	**+**	**2931**	+	or	-	171	-	34.13	-	172.4	-	223	-	12.99	-	299.05
Goat milk S2	+	12458.5	+	2521	+	or	-	2	-	9.2	-	153.8	-	261.5	-	16.12	**+**	**409.15**
Sheep milk S1	+	20000	+	2766	+	or	**+**	**722.5**	-	162	**+**	**1006.95**	-	93.5	-	14.92	-	379.05
Sheep milk S2	+	20000	+	2851	+	or	-	14	-	25.37	-	83.2	-	22.93	-	90	-	180.2
Camel milk S1	+	20000	+	781	+	10660	**+**	**2255.5**	-	180.1	**+**	**6102.5**	**+**	**1921.5**	-	186.3	**+**	**7736.5**
Camel milk S2	+	4249	+	494	+	4386	**+**	**1754.5**	-	146	**+**	**4879.5**	**+**	**1911**	-	63.71	**+**	**2537.1**
Donkey milk S1	+	1251.5	**-**	**111**	+	2167.9	-	9	-	20.97	-	66.65	-	15	-	11.67	-	100.9
Donkey milk S2	+	2075.5	**-**	**93**	+	2060.4	-	7.5	-	7.52	-	16.55	-	22	-	30.6	-	123.45
Buffalo milk S1	+	3631.5	+	5304	+	or	-	11	-	8.11	-	46.2	-	3	-	22.87	-	54.95
Buffalo milk S2	+	13495	+	5325	+	or	-	85	-	19.52	-	209.6	-	13.5	-	46.12	-	37.25

or: over ranged. In bold, the results that are considered not correct, according to currently established limits.

**Table 5 biotech-11-00039-t005:** Comparative results of all ALP tests of non-cow milk samples.

	Qualitative Tests			Quantitative Tests		
	A	B	C	D	E	F
Goat milk S1 raw	+	- *	- *	+	+	+
Goat milk S1 HTST	~ *	-	-	-	-	-
Goat milk S1 LTLT	~ *	-	-	-	-	-
Goat milk S2 raw	+	+	+	+	+	+
Goat milk S2 HTST	~ *	-	-	-	-	-
Goat milk S2 LTLT	~ *	-	-	-	-	+ *
Sheep milk S1 raw	+	+	+	+	+	+
Sheep milk S1 HTST	~ *	-	-	+ *	-	+ *
Sheep milk S1 LTLT	-	-	-	-	-	-
Sheep milk S2 raw	+	+	+	+	+	+
Sheep milk S2 HTST	-	-	-	-	-	-
Sheep milk S2 LTLT	-	-	-	-	-	-
Camel milk S1 raw	+	+	+	+	+	+
Camel milk S1 HTST	+ *	-	-	+ *	-	+ *
Camel milk S1 LTLT	+ *	-	-	+ *	-	+ *
Camel milk S2 raw	+	+	- *	+	+	+
Camel milk S2 HTST	+ *	-	-	+ *	-	+ *
Camel milk S2 LTLT	+ *	-	-	+ *	-	+ *
Donkey milk S1 raw	+	- *	- *	+	- *	+
Donkey milk S1 HTST	-	-	-	-	-	-
Donkey milk S1 LTLT	~ *	-	-	-	-	-
Donkey milk S2 raw	+	- *	- *	+	- *	+
Donkey milk S2 HTST	-	-	-	-	-	-
Donkey milk S2 LTLT	~ *	-	-	-	-	-
Buffalo milk S1 raw	+	+	+	+	+	+
Buffalo milk S1 HTST	-	-	-	-	-	-
Buffalo milk S1 LTLT	-	-	-	-	-	-
Buffalo milk S2 raw	+	+	+	+	+	+
Buffalo milk S2 HTST	-	-	-	-	-	-
Buffalo milk S2 LTLT	-	-	-	-	-	-

* results that are considered not correct, according to currently established limits and milk sample status.

## Data Availability

Not applicable.

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
