# Peer review of "Preliminary Results on the Comparative Evaluation of Alkaline Phosphatase Commercial Tests Efficiency in Non-Cow Milk Pasteurization"

_biotech, 2022, doi:10.3390/biotech11030039_

Round 1

Reviewer 1 Report

The manuscript “Preliminary results on the comparative evaluation of alkaline phosphatase commercial tests efficiency in non-cow milk pasteurization” belongs to the area of food safety. Verifying the safe level of pasteurization is necessary for ensuring the food safety of milk products, so it is important to verify the correctness of the test procedures which are used to control the quality of pasteurization. These procedures are generally developed for the cow-milk, so it is necessary to check if they are valid for non-cow milk.

The typical tests rely on the evaluation of alkaline phosphatase activity, which itself varies in different species, thus the adequate performance of the test developed for cow milk is not given for other animals’ milk.

The authors have analyzed the performance of three qualitative and three quantitative test that rely on the detection of ALP activity to check the milk status. The studied milk came from goat, sheep, camel, donkey, and buffalo. Out of all combinations studied, about 17% came erroneous, with both false negative and false positive results. It appears only the Fluorophos test correctly determines the quality of milk for all studied species.

The described results illustrate the need to further develop the methods for analysis of the non-cow milk and raise important objections over the application of the tests protocols developed for cow milk in non-cow species.

Overall, the article is well written and can be accepted after minor correction.

The list of the issues I found is presented below:

Line 18: “thermal inactivation of the kinetics” it seems that the word “kinetics” is redundant, do the authors mean “thermal inactivation of an endogenous enzyme”?

Lines 44-46: while the amount of released phenol is a measure of ALP activity, I think the way the sentence is presented may confuse some people. Also, the link [3] does not seem to describe the changes of the ALP activity during lactation in sheep directly.

Line 78: again, the link to [3] does not seem correct, as the paper in question itself quotes other studies on the proposal for different legal limits.

Lines 109-111: No link given for the buffalo milk study.

Line 165: “a-cyano-4-hydroxycinnamic acid” alpha symbol is not displayed correctly.

Line 177: dot is missing at the end of the sentence.

Lines 180-181 (also 201): “The technical characteristics of each test, briefly refer to the following:” Perhaps “For” is missing at the start of the sentence?

Line 188: “shaked”, must be “shaken”.

Line 207: unnecessary parenthesis.

Line 213: comma seems inappropriate.

Author Response

Dear Reviewer,

Thank you for your valuable comments, we agree with all of them, please see below a point-to-point response:

Line 18: “thermal inactivation of the kinetics” it seems that the word “kinetics” is redundant, do the authors mean “thermal inactivation of an endogenous enzyme”?

Response: We agree with the comment, the phrase has been changed accordingly please see line 18.

Lines 44-46: while the amount of released phenol is a measure of ALP activity, I think the way the sentence is presented may confuse some people. Also, the link [3] does not seem to describe the changes of the ALP activity during lactation in sheep directly.

Response: We agree with the comment, the correct citation is no12 (IZSLT, 2020), it has been moved to no3. Klotz et al has been moved to no 9 and citations 10,11,12 have been changed accordingly.

Line 78: again, the link to [3] does not seem correct, as the paper in question itself quotes other studies on the proposal for different legal limits.

Response: We agree please see previous response, citations have been changed accordingly.

Lines 109-111: No link given for the buffalo milk study.

Response: We agree, the relevant citation has been added, please see citation no15. Following citations have been changed accordingly

Line 165: “a-cyano-4-hydroxycinnamic acid” alpha symbol is not displayed correctly.

Response: We agree, symbol has been cjanged to α. 

Line 177: dot is missing at the end of the sentence.

Response: We agree, the dot has been added.

Lines 180-181 (also 201): “The technical characteristics of each test, briefly refer to the following:” Perhaps “For” is missing at the start of the sentence?

Response: We agree, "For" has been added in both cases.

Line 188: “shaked”, must be “shaken”.

Response: We agree, the word is changed accordingly.

Line 207: unnecessary parenthesis.

Response: We agree, the parenthesis was removed.

Line 213: comma seems inappropriate.

Response: We agree, the comma was removed.

Reviewer 2 Report

In this manuscript, the authors evaluate the efficiency of commercially available ALP test kits in non-cow milk. This evaluation is necessary to understand ALP limitation in various species type of milk (non-cow).

In this study,

·       Authors have used milk from five species - sheep, goat, buffalo, donkey and camel.

·       There are three sample conditions: raw-milk, high temperature short time (72 °C, 15sec) pasteurized milk, and low temperature long time (63 °C, 30 min) pasteurized milk

·       To confirm thermal effect on microorganism samples were also analyzed microbiologically using standard ISO methods and MALDI-TOF MS

·       The qualitative and quantitative analysis are done with three kits, respectively:

A.    Lactognost (HEYL),

B.    Phosphatesmo (MA-CHEREY-NAGEL GmbH & Co. KG) and

C.    Lactopast Biomedix (MenidiMedica)

D.    ZymoSnap ALP (Hygiena),

E.    PasLite test (Charm Sciences INC) and

F.     Fluorophos (Advanced Instruments)

In the case of camel milk, this study agrees with previous observation that ALP assay cannot be used as pasteurization indicator.

Although the result presented in this study is not surprising, in light of the earlier claims, its experimental evaluation is important.

Overall, the project is well conceived, the data are convincing, and the discussion is brief but complete. The manuscript is suitable for publishing in BioTech after making the following minor correction:

A couple of comments follow:

·       Readers would want cow milk as a positive control to evaluate kits. Although ALP activity in buffalo milk similar to the cow milk and authors have used buffalo milk. Thus, buffalo milk should be mentioned as a positive control.

·       There are few typos and inconsistencies:

o   Line# 143,146,147, 148 either keep the space before units ‘°C’, ‘ml’ & ‘min’ or not.

o   Table-1, TVC values for raw milk in Donkey and buffalo are merged with S1 and S2 and gives an illusional value.

o   This review should to be cited:  J. Dairy Sci.  93 :5538–5551 doi:  10.3168/jds.2010-3400 

Author Response

Dear Reviewer,

Thank you for your valuable comments, we agree with all of them. Please see below a point-to-point response:

Readers would want cow milk as a positive control to evaluate kits. Although ALP activity in buffalo milk similar to the cow milk and authors have used buffalo milk. Thus, buffalo milk should be mentioned as a positive control.

Response: We agree with the reviewer, the following sentence was added: "As the activity of ALP in buffalo milk is similar to cow milk, this type of milk served as a positive control for all tests." Please see line 178.

·       There are few typos and inconsistencies:

o   Line# 143,146,147, 148 either keep the space before units ‘°C’, ‘ml’ & ‘min’ or not.

Response: We agree, the space was kept in all cases.

o   Table-1, TVC values for raw milk in Donkey and buffalo are merged with S1 and S2 and gives an illusional value.

Response: We agree, we changed the cells width accordingly.

o   This review should to be cited:  J. Dairy Sci.  93 :5538–5551 doi:  10.3168/jds.2010-3400 

Response: We agree, please see citation no 20.